

# Numerical Models for Monitoring and Forecasting Ocean Ecosystems: a short description of present status

Simone Libralato[1]

[1]Istituto Nazionale di Oceanografia e di Geofisica Sperimentale, Department of Oceanography, Trieste, Italy

*Correspondence to*: Simone Libralato (slibralato@ogs.it)

**Abstract.** Understanding and managing marine ecosystems under potential stress from human activities or climate change
requires the development of models with different degree of sophistication in order to be capable of predicting changes in living components and environmental variables. Recent advances in ecosystem modelling are the focus of this paper, which reviews numerical approaches to analyse the characteristics of marine conditions in terms of typical units, i.e., individuals, populations, communities and ecosystems. In particular, it examines the current classification of numerical models of increasing complexity – from individuals and population and stock assessment models to models representing the whole
ecosystem by covering all trophic levels – and presents examples and their operational maturity, finally demonstrating their use for supporting marine resource management, conservation, planning and mitigation actions.

## 1 Introduction

In recent decades, a variety of numerical models have been developed to predict the effects of various environmental changes and human impacts on marine biological resources. A comprehensive analysis is challenging, although models can be mapped
in terms of their main scope and distinguishing approaches that can incorporate age structure, environmental factors, represent trophic interactions, and spatial structure (Hollowed et al., 2000; Plaganyi, 2007). Based on the above characteristics, numerical models can be divided into six broad classes:

- Bioenergetic models representing the processes related to growth, respiration, excretion of an individual;
- population and stock assessment models (typically for single species without trophic interactions and possibly age-
structured);
- connectivity models (considering propagules dispersal, larval cycle, spatial structures, and environmental factors);
- species distribution models (statistical models based on representation of spatial environmental variables and biota);
- minimal realistic models (typically age-structured, with a few species trophic interactions);
- whole ecosystem models (typically covering all trophic levels and based on trophic interactions, which may include
size structure and spatial variation).

These five classes of models are reviewed in the following sections, considering also available syntheses and reviews (e.g., Plaganyi, 2007; Stock et al., 2011; Itoh et al., 2018). For each class of models some examples are reported and their characteristics in terms of typical units, elemental structure, number of species, representation of trophic interactions are reported in Table 1. It also contains synthetic information on primary model focus and main output, as well as if each model

is operational or not.

## 2 Bioenergetic models

A bioenergetic model is any mechanicistic model describing how individuals take energy from the environment and allocate it to different processes (Kooijman, 2010; Sibly et al., 2013). Bioenergetic models are typically used for representing the growth of the individual, while accounting for respirations, excretions and other losses. Energy intake can be a model input or

output, depending on whether it's modeled dynamically or derived from energy requirements (Pirotta et al., 2022). Acquisition and allocation can vary based on the individual's state and environmental conditions (e.g., Libralato and Solidoro, 2009; Nisbet et al., 2012)

Indeed, bioenergetic models can account of external oceanographic conditions influencing uptakes, such as light, nutrients and temperature for autotrophs (e.g., Bocci et al., 1997) or food availability and temperature for heterotrophs (e.g., Libralato and

Solidoro, 2009), while losses are usually related to temperature and internal conditions (Koojman, 2010). Bioenergetic models can also consider explicitly the gonadic development and egg release (Pastres et al., 2000).

Traditional bioenergetic models describe energy intake from feeding and its allocation to maintenance, activity, growth, reproduction, and excretion. These models are advantageous due to their clear empirical interpretation and measurable units but tend to be parameter-heavy and difficult to generalize across species. A widely used approach for fish and invertebrates is

represented by the Dynamic Energy Budget (DEB, Koojiman, 2010) which is characterized by an explicit representation of somatic, gonadic and storage tissues. Dynamic Energy Budget (DEB) theory offers a more general approach by using mass-energy balance principles to link sub-organismal processes with overall organismal performance (Koojiman, 2010; Nisbet et al., 2012). However, this generality leads to abstract concepts that are more challenging to measure empirically (Pirotta et al., 2022). The presence of the storage in DEB allow representation of delayed use of energy in the individual development. DEB

has been developed into a theory for scaling the parameters for all life cycles of the individual (from eggs to larvae to juveniles and adults) and setting parameters for a large number of marine species (Nisbet et al., 2012), thus it has a maturity for being used routinely and adapted to operational applications.

## 3 Population and stock assessment models

Various types of numerical models of single populations are used worldwide to support management by determining the

population at sea and the current status of exploited marine populations, thus providing insight for management. Such stock





assessment models typically represent the biomass or abundance of individual species, are routinely used by management agencies, and include probability models to incorporate various sources of observational data (Maunder and Punt, 2013). In cases where stock assessments are based on limited observations, i.e., no catch by age or size, surplus production models are used. The general underlying assumption of these models is a theta-logistic function for the evolution of biomass (B; but it can

also be applied to the number of individuals) over time:

$$\frac{dB}{dt} = \frac{r}{n-1} B_t \left( 1 - \left[ \frac{B_t}{K} \right]^{\theta-1} \right) - F_t B_t \quad (Eq.1)$$

where F is the fishing mortality, so $F_t B_t$ corresponds to the catches Ct, while r is the population growth rate, and K is the capabilities of the system to support the population (through living space, habitat, food, etc.), and generally called as carrying capacity. The parameter Ө allows generalization of the equation (in the case of Ө=2, the classical logistic curve is obtained).

Several packages use the surplus production model as principal approach, are used routinely to perform stock assessment and are including several useful diagnostics. Among them the most used are the CMSY (Froese et al., 2023) and the SPiCT (Pedersen and Berg, 2017) models. CMSY uses a time series of catch data and fishing effort to estimate parameters, reconstruct biomass and establish reference points using a Bayesian approach (Froese et al., 2023). The Stochastic surplus Production model in Continuous Time (SPiCT) provides estimates of exploitable biomass and fishing mortality at any point in time from

data collected at arbitrary and possibly irregular intervals (Pedersen and Berg, 2017). The model allows the inclusion of prior distributions for parameters that are difficult to estimate such as growth rate and carrying capacity. SPiCT is available as an R package (R Core Team 2015) in the online GitHub repository: https://github.com/mawp/spict.

Surplus production models are simplistic representation of the population that is lumped with no size and/or age details. More sophisticated approaches (such as SS3, a4a, XSA, etc.) are used when data by age or size classes are available for the exploited

population (catch-at-age or catch-at-length models; Maunder and Punt, 2013). These stock assessment models reconstruct the number of individuals in cohorts based on catch and natural mortality by age class, as well as information on species growth, fecundity, and fisheries selectivity (Methot and Wetzel, 2013). The basic dynamics are described by the number of individuals N at time t and age a, as in the following:

$$N = R_t + \left( N_{t-1,a=1} e^{-\frac{M}{2}} - C_{t-1,a=1} \right) e^{-\frac{M}{2}} + \left( N_{t-1,a=2} e^{-\frac{M}{2}} - C_{t-1,a=2} \right) e^{-\frac{M}{2}} + \cdots \left( N_{t-1,x-1} e^{-\frac{M}{2}} - C_{t-1,x-1} \right) e^{-\frac{M}{2}}$$

$$+ \left( N_{t-1,x} e^{-\frac{M}{2}} - C_{t-1,x} \right) e^{-\frac{M}{2}} \quad (Eq.2)$$

where each year the population comprises all age classes from the new juvenile individuals entering the population as recruits the same year ( R, age 0 ), all age classes a, from 1 up to the oldest age modelled (age x) surviving from the year before. The number of individuals are decreasing through time on the basis of catches C at age and time, and assuming instantaneous

natural mortality M.





Typically, these models report juvenile recruitment R as a function of a combination of fecundity by age class estimated from data (Stock et al., 2011). This class of models includes, for example, the a4a tool (assessment for all, Jadim et al., 2014), a modeling framework for fitting statistical age-structured fishery models using nonlinear statistical submodels. The submodels can include linear functions of age and year, smoothing splines with fixed degrees of freedom that vary with age and/or year

and environmental indicators as covariates. The tool a4a is implemented in R Fishery Library and includes the optimization procedure. Stock synthesis (SS3; Anderson et al., 2014) is the most widely used catch-at-age stock assessment model that can incorporate age or length composition information from surveys, abundance indices, multi-gear effort, selectivity, and spatial data in the most recent and advanced applications (e.g., Punt, 2019; Privitera-Johnson et al., 2022). In all cases, projections of stock assessment models are generally made for annual to decadal time periods. Catch or effort limitation scenarios can be

used to estimate biological reference points for management decisions (indicators based on maximum sustainable yield). Although in most cases, stock assessment models are not spatially explicit and do not consider explicitly oceanographic forcings they are routinely used in formal assessments for management and might be considered as ready for operational oceanographic applications.

**4 Connectivity models**

The distribution and survival of small eggs and larvae of marine fishes and invertebrates, as well as propagules of algae and seagrass' seeds are advected and thus are strongly influenced by currents, which can disperse individuals both near spawning sites and in distant areas (Cowen et al., 2007). Therefore, biophysical dispersal (advection, diffusion, and migratory behavior of organisms) is fundamental to explaining population dynamics and connectivity (Cowen et al., 2009). Numerical models are used to quantitatively integrate the large spatial and temporal variability of oceanographic processes (physical connectivity)

with processes inherent in the biology of marine organisms (life history traits) to investigate connectivity between and within populations and also across larval stages (Gawarkiewicz et al., 2007; Melaku Canu et al., 2021). Connectivity models typically use offline physical parameters (velocity, density, temperature, and salinity) obtained from hydrodynamic models and estimate the distribution of organisms: since in most of the cases living organisms have negligible influences on physical oceanographic processes parameters, modeling the biophysical dispersion offline from the hydrodynamic models is considered a reliable

strategy also considering time evolutions. The advection–diffusion–reaction equation is typically used for biomass distribution (e.g., Sibert et al., 1999; Faugeras and Maury, 2005), while Lagrangian approaches are used to track particles and thus distribute individuals (e.g., Laurent et al., 2020). These approaches take into account life history traits such as growth, mortality and the behavior of target organisms in terms of seasonal variability, spawning sites, vertical movement and settlement preferences [e.g., Melaku Canu et al., 2021; Paris et al., 2013; Lett et al., 2008]. Connected with oceanographic variables and spatially

explicit these models however, appear less mature for operational applications.



## 5 Species distribution models

Species distribution models (SDM, also called habitat suitability models) are statistical models that predict the occurrence, abundance, or biomass of organisms using geoposition, biotic and environmental data (Brodie et al., 2020). Particularly useful when applied to standardized, scientific monitoring and surveys of biotic data, these approaches can also exploit publicly available datasets (e.g., OBIS, www.obis.org; GBIF, www.gbif.org). SDMs are implemented using various approaches, including linear models (LM), generalized linear models (GLM), generalized additive models (GAM) (Melo-Merino et al., 2020; Maravelias et al., 2003), machine learning methods such as random forest (RF, Breiman et al., 2018) or artificial neural networks (ANN), and maximum entropy (Jones et al., 2012; Pittman and Brown, 2011; Reiss et al., 2011). The inclusion of physical and biogeochemical oceanographic covariates, which can have direct and indirect effects on species distributions, can improve the capabilities of SDMs to explain observed biotic data compared to using only geopositional variables (Panzeri et al., 2021; Thorson et al., 2015). Recent advances include combining the approaches into an ensemble (Jones et al., 2012) and including multiple species as covariates into the so called Joint-Species Distribution Models (Pollock et al., 2015). These classes of SDMs are increasingly being used to describe current and future distributions of exploited and endangered species, identify hotspots, map essential fish habitat, support conservation development, and feed other ecosystem models (Jones et al., 2012; Colloca et al., 2015; Grüss et al., 2014; Dolder et al., 2018).

The Dynamic Bioclimate Envelope Model (DBEM) estimates species distributions based on environmental preferences and considers population dynamics and dispersal (Cheung et al., 2009). The DBEM makes predictions of future envelopes using physical and biogeochemical data from oceanographic models and also considers the response of organisms to natural/anthropogenic environmental changes such as growth, mortality, larval dispersal, and migration (Cheung et al., 2013). In general SDMs are widely applied, and although at the moment they are not used operationally, they can be easily implemented within an operational chain.

## 6 Minimal realistic models

Dynamic multispecies models or Minimal Realistic Models (MRM, Punt and Butterworth, 1995) are approaches that represent a limited number of species (usually less than 10 species) that have important interactions with a target species. The MRMs often represent an evolution of single species stock assessment models: for example, Multispecies Virtual Population Analysis (MSVPA) is an extension of virtual population analysis (Gislason, 1999), while GADGET (Globally applicable Area-Disaggregated General Ecosystem Toolbox) is an extension of stock synthesis in the multispecies framework, where populations can be partitioned by species, size classes, age groups, areas, and time steps (e.g., Andonegi et al., 2011). In particular, GADGET is flexible, allowing easy addition/replacement of alternative model components for biological processes such as growth, maturation, and predator-prey interactions representing some species in age classes. GADGET provides estimates of population dynamics under technical and biological interactions with the ability to use different growth functions and fitness functions (Plaganyi, 2007).



MICE ("Models of intermediate complexity for ecosystem assessment"; Plagányi et al., 2014) are developed considering the specific problem and data availability. MICE represents a limited number of populations (usually 10) exposed to fisheries or
anthropogenic interactions and includes ecological processes (Angelini et al., 2016). These models have different levels of detail for the species represented: MICE can simultaneously represent focal populations in age-structured classes, while others take a surplus production approach (Morello et al., 2014). MICE can be a fairly complex but flexible tool that overcomes the many complexities of whole ecosystem models and is useful for providing tactical advice for focal species management (Plagányi et al., 2014). Spatial Environmental POpulation Dynamics Model (SEAPODYM) is a two-dimensional coupled
physical-biological model originally developed for tropical tunas in the Pacific (Lehodey et al., 2003). SEAPODYM includes an age-structured population model for tuna species and a movement model based on a diffusion-advection equation such that swimming behavior is modeled as a function of habitat quality (sea surface temperature (SST), ocean currents, and primary production) predicted from oceanographic models (Lehodey et al., 2015). This model describes spatial structures that are essential to account for the distribution of fishing effort, swimming behavior, and environmental variations typically
determined by ocean circulation models or derived from satellites (Lehodey et al., 2015; Senina et al., 2020).

**7 Whole ecosystem models**

Whole ecosystem models (WEM) are designed to represent all trophic levels in an ecosystem, from primary producers to top predators. Thus, WEMs typically use a very large set of data collected from a variety of disciplines, including results from oceanographic models and stock assessments (e.g., Agnetta et al., 2022).
The main distinguishing feature between the different WEM is the way in which the ecosystem is described: i) in flexible compartments representing species, ecologically meaningful groups of species, or size- and age-structured populations, such as Ecopath with Ecosim (hereafter EwE, Christensen and Walters, 2004) and ATLANTIS (Fulton et al., 2005); ii) in size-structured communities, typically benthic and pelagic communities, such as Osmose (Shin and Cury, 2001), Feisty (Petrik et al., 2019), and DBEM (Blanchard et al., 2009), for example; iii) in a mixture of size-structured communities (typically pelagic,
mesopelagic migratory, and non-migratory) and age-structured species as in Apecosm (Maury, 2010); iv) the ecosystem is described by dynamic spectra of trophic levels as in Ecotroph (Gasche and Gascuel, 2013). All these models are based on biomass and consider rules such as biomass conservation.
EwE, undoubtedly the most widely used WEM, is a free, general model for whole ecosystems that have been developed over 35 years (Christensen and Walters, 2004) and has been used to analyze past and future impacts of fisheries, nutrient inputs,
invasive species, and climate (e.g., Heymans et al., 2014; Libralato et al., 2015; Serpetti et al., 2017; Piroddi et al., 2021). It consists of three different interconnected main modules, i) a static mass-balanced ecosystem network (Ecopath, Christensen and Pauly, 1992), ii) a temporally dynamic simulation module (Ecosim, Walters et al., 2000), and iii) a spatially and temporally dynamic module (Ecospace, Walters et al., 1999). EwE contains a large number of additional modules for calibration, uncertainty analysis, calculation of indicators, and simulation of pollutant dynamics (Steenbeek et al., 2016). Recent advances



allow the direct embedding of two-dimensional monthly results from oceanographic physical-biogeochemical models (Steenbeek et al., 2013).

ATLANTIS spatially resolves the full trophic spectrum of ecosystem types, including age-structured formulations for high trophic levels, potentially in multiple vertical layers (Fulton et al., 2011). ATLANTIS includes a nutrient pool formulation that can be used to test effects such as nutrient inputs (Audzijonyte et al., 2019). The model has been used for site-specific analyses

and to examine general aspects of fishery's impacts on fish communities (Link et al., 2010).

OSMOSE (Objected-oriented Simulator of Marine ecOSystems Exploitation; Shin and Cury, 2004) is an individual-based ecosystem model that simulates size-based communities on a 2-D spatial cell grid and can be coupled with a planktonic ecosystem model (Travers et al., 2010). The model has been used to study the effects of various aspects of fisheries on the food web (e.g., Shin and Cury, 20019.

These models WEM are increasingly being used to address the need for holistic ecosystem approaches, and their framework is often used to answer strategic medium-term questions related to management strategies, fisheries issues, and climate or environmcental change (e.g., Tittensor et al., 2021). Notably, WEM can be coupled with other classes of models (population dynamic, SDM, connectivity models), as well with biogeochemical models.

**Conclusions**

A large set of models exist that were developed for representing individuals, populations, communities and whole ecosystem. These models have been developed for specific objectives that embrace many issues important for society, i.e., from effects of climate change, pollution, nutrient enrichment, fisheries etc. The numerical approaches analysed here have characteristic spatio-temporal resolution (Table 1) generally decreasing when moving from individual species models to whole ecosystem models. Moreover, while increasing complexity with MRM and WEM there is general improvement of realism but also lower

accuracies. Overall the first set of approaches (bioenergetic and population models) are more adapted for tactical analyses while especially the WEM are at the moment considered useful especially in strategic analyses (see Table 1). Although very few of the reviewed approaches are currently used in operational approaches (e.g., stock assessments), all the tools have great potentials for becoming operationally used for analyse ecosystem dynamics and make useful scenarios, on a very wide range of issues and management actions that might be eventually prioritized.

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

**Competing interests**

The contact author has declared that none of the authors has any competing interests.

**Data and/or code availability**

Not applicable.

**Authors contribution**

Not Applicable

**Acknowledgements**

A draft version of this paper benefited from comments and suggestions received by Vinko Bandelj (OGS), Elisa Donati (UNITS and OGS), Celia Laurent (OGS), Ivano Vascotto (OGS), Damiano Baldan (OGS), Fabrizio Gianni (OGS) and Davide Agnetta (OGS).


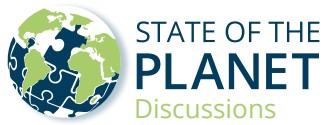
STATE OF THE PLANET Discussions

**Table 1: Main characteristics of widely used numerical models for marine biological resources**

**Bioenergetic models**

| Model | Name | Elemental structure | Model Units | Time units | Spatial structure | Number of species | Trophic interactions | Primary model focus, output | Operational | Physical/biogeochemical processes |
|---|---|---|---|---|---|---|---|---|---|---|
| DEB | Dynamic Energy Budget | Individual | Individual weight (gww, gC or others) or length | day | No | 1 | No | growth | Potentially | Yes used as forcings (temperature, light, food, nutrients) |

**Population and stock assessment models**

| Model | Name | Elemental structure | Model Units | Time units | Spatial structure | Number of species | Trophic interactions | Primary model focus, output | Operational | Physical/biogeochemical processes |
|---|---|---|---|---|---|---|---|---|---|---|
| Spict | stochastic surplus production model in continuous time | Surplus prodution | Biomass | Year | No | 1 | No | Biological reference points | yes | No |
| CMSY | Catches at Maximum sustainable Yield | Surplus prodution | (tonn) | Year | No | 1 | No | Biological reference points | Yes | No |
| A4a | All for all | Catch-at-age | Biomass (tonn) | Year | No | 1 | No | Biological reference points | Yes | No |


STATE OF THE PLANET
Discussions

| Model | Name | Element al structure | Model Units | Time units | Spatial structure | Number of species | Trophic interactions | Primary model focus, output | Operational | Physical/biogeochemical processes |
|---|---|---|---|---|---|---|---|---|---|---|
| SS3 | Stock Synthesis | Catch-at-age | Number Individuals; biomass (ton) | Year | Potentially yes | 1 | No | Biological reference points | Yes | Potentially yes |
| VPA | Virtual population analysis | Catch-at-age | Number Individuals; biomass (ton) | Year | no | 1 | No | Biological reference points | no | No |

**Connectivity models**

| Model | Name | Element al structure | Model Units | Time units | Spatial structure | Number of species | Trophic interactions | Primary model focus, output | Operational | Physical/biogeochemical processes |
|---|---|---|---|---|---|---|---|---|---|---|
| LTRANS | Lagrangian Transport | Agents (super individuals) | Number of individuals | days | Yes | Typically one species | No | Distribution of species and connectivity among sites | No | Yes |
| IBM/ABM | Individual-based and Agent Based Models | Individual | Biomass | days | Yes | Typically a few species | Efficient predator | Ecosystem effects on target population and connectivity | No | Yes |

**Species Distribution Models**

| Model | Name | Element al structure | Model Units | Time units | Spatial structure | Number of species | Trophic interactions | Primary model focus, output | Operational | Physical/biogeochemical processes |
|---|---|---|---|---|---|---|---|---|---|---|





| Model | Name | Elemental structure | Model Units | Time units | Spatial structure | Number of species | Trophic interactions | Primary model focus, output | Operational | Physical/biogeochemical processes |
|---|---|---|---|---|---|---|---|---|---|---|
| Ensemble of SDM | Ensemble of Species Distribution models | Species abundance, presence or biomass | Number of individuals or weight per unit surface or presence/absence | Month, year, climatology | Yes | 1 | No | Species distribution; essential fish habitats | Potentially | Environmental factors can be included |
| Joint-SDM | Joint Species Distribution models | Species abundance, presence or biomass | Number of individuals or weight per unit surface | Month, year | Yes | A few species | implicit | Distribution of target species | No | Environmental factors can be included |
| DEBM | Dynamic Bioclimate Envelope Model | Species biomass | biomass | year | Yes | Several species | No | Distribution of multiple species | No | Yes included for developing the bioenvelope |
| **Minimal Realistic models** | | | | | | | | | | |
| GADGET | Globally applicable Area Disaggregated General Ecosystem Toolbox (derived from BORMICON) | Population in age structure | Biomass derived from population size structure | Year | Yes, can be included | Typically 3-4 species | Yes, suitability-based, flexible | Ecosystem effects on target population; yearly biomass | potentially | Can be coupled with physical-biogeochemical model |





**Whole Ecosystem Models**

| Model | Name | Elemental structure | Model Units | Time units | Spatial structure | Number of species | Trophic interactions | Primary model focus, output | Operational | Physical/biogeochemical processes |
|---|---|---|---|---|---|---|---|---|---|---|
| MSVPA and MSFOR | Multi-species Virtual population Analysis and multi-species Forecasting Model | Populations in age structure | Numbers at age; Biomass | Year | No | Typically 3-4 species | Yes; Suitability-based; Efficient predator | Ecosystem effects on target population; yearly biomass | No | Not usually included |
| MICE | Model of Intermediate Complexity for Ecosystem assessments | Populations in surplus production and age structure | Numbers at age, Biomass | Year | No | Typically 6-7 species | Efficient predator | Dynamics of focal species and their predators or preys | Potentially | Environmental effects can be included |
| SEAPODYM | Spatial Ecosystem, and population Dynamics Model | Populations in age structure | Biomass | Year | Yes | Typically 3-4 species | Efficient predator | Ecosystem effects on target population | Yes (tuna) | Can be coupled with physical-biogeochemical model |
| | Commission for the Conservation of Antarctic Marine Living Resources | Functional group approach | | | | Limited number of HTL groups | | Effects in both directions | No | |
| ERSEM II | | | Nutrient | month | Yes | | Type II | | No | Yes, detailed |





| | | | | | | | | | | |
|---|---|---|---|---|---|---|---|---|---|---|
| ATLANTIS | Atlantis | Population s in age structure | Nutrient | month | Yes | Can be a very large number ; typicall y order 40 | Flexible, Type II, type III or other | Effects of ecosyste m and fisheries in both directions ; yearly outputs | No | Yes, detailed |
| EwE | Ecopath with Ecosim | Function al group approach ; Populatio ns also in age structure | Biomass, Nutrient | month | Yes (ECOSPA CE) | Can be a very large number ; typicall y order 40 | Foraging arena, flexible approach | Effects of ecosyste m and fisheries in both directions ; yearly outputs | No | Included as off-line coupling |
| OSMOSE | Object-oriented Simulator of marine ecosystem Exploitatio n | Size spectra approach | Biomass at different levels of aggregation | year | Yes | Large number of species | Efficient predator but can starve | Multispec ies dynamics | No | Included as off-line coupling |
| FEISTY | | Size spectra approach | Biomass at different levels of aggregation | year | Yes | Large number of species | Flexible approach | Multispec ies dynamics | No | Included as off-line coupling |
| Apecosm | | | | | | | | | | |