# Peer review of "Numerical Models for Monitoring and Forecasting Ocean Ecosystems: a short description of present status"

_State of the Planet, 2024_

## Author Response (AR2)

**Point to point response: SP-2024-42**

In **bold** the comments provided by the three reviewers (RC1, RC2 and RC3). In *italics* my answers.

**RC1**

**The paper provides an overview of six classes of marine ecosystem models and evaluates their current status in operational applications. I acknowledge that it's a daunting task to review these many models in such limited space, and I appreciate the author's efforts. However, I found some aspects of the paper challenging.**

1. **I wonder who the targeted audience for this review is. I'm afraid that the current version is not quite accessible to the wider audience, partiallydue to the use of jargon and acronyms—some of which are not explained—and by lengthy sentences from time to time. I have a background in ocean ecosystem modeling, but I have to admit that it's not always easy to follow the description of all models. If the review is intended for a wider audience, I think it could be a bit more elementary.**

*R1_1 - The paper intends to provide a structured synthesis of models applied to marine higher trophic levels (basically from zooplankton to top predators and fisheries) that can be connected with lower trophic level models (physics and biogeochemistry). The lower trophic level models currently provide operational products in Copernicus Marine Service and their potential to be connected with high trophic level models open wide spectrum of new CMS products. The objective is to cover most of the approaches from single individual to population to multispecies but necessarily the work cannot be exhaustive. Conversely, the work provides some classification and examples of most used high trophic level tools with some indication of their potential use for an operational connection with lower trophic level models. A few statements were added in the introduction in order to clarify objectives and express the above issues and needs. Given the broad spectrum of approaches, that have been developed in connection to specific scientific objectives, possibly the paper resulted in using sector-specific gergon also for the sake of brevity. In the revised version I reduced at maximum the use of acronyms, parameters and gergon. An English revision helped making the document more accessible from a large audience, by harmonizing the language in all 6 classes of models.*

2. **The Introduction indicates that there are "available syntheses and reviews". How does this short description complement those previous works? A brief summary highlighting the specific focus of the previous reviews in relation to this one would help readers identify which resources best align with their interests.**

*R1_2- The work collates some information from different reviews and does not intend to substitute to them. Reviews are complete each on one or two classes while the work intends to synthetically bridge across more classes and approaches. The work does not pretends to be exhaustive but examples of tools for each class might shed light on their usage for operational coupling with lower trophic level models. In the revised version clear objectives are stated in a couple of sentences.*

3. **As a follow-up comment, given the limited space available for detailed information on each class of model, it would be beneficial if the author could recommend key references for further reading.**

*R1_3- For each session it is now clearly reported a key review that readers can use to have deeper insights.*

4. **The criteria used to assess the readiness or maturity of each class of model for operational applications could be more explicitly defined. Is it based on the model complexity, computation cost, or technical challenges in implementation?**

*R1_4- This is a key element that was too simplified before and in need of more explanations. Readiness and maturity of each model was subjectively elaborated on the basis of its current application and information on the tool. Notably, tools routinely applied in fisheries management, for example, were considered more mature because the code is publicly available and documented and input and output test cases are developed and accessible. The most mature models also embed routines for assessing model performances, diagnostics and are used by a community of developers that can provide support, updates and advancement. In this revised version I made an effort to distinguish between maturity and readiness to be used for operational purposes. Maturity is now explicitly referring to the above points (code availability, documentation, test cases, diagnostics for model performances and reference community), thus explaining an assessment which is subjective but quite well circumstantiated. Readiness for operational purposes, will be explicit and would be defined on the basis of existing knowledge on possible connection of the model to physical and biogeochemical spatio-temporal models. Existence of such applications, even if scarce, might shed light on the difficulties in connecting (one way or two way) with low trophic level models.  Readiness for operational purposes might be considered more tentative and less precise, because it is more difficult to define in a very objective way also for the potentially very sparse  application. Both maturity and readiness definition are now proposed in the introduction, used in each sections to assess modelling examples and used in Table.*

5. **Given the limited space, the description of each model class is brief. The only exception is the "3 Population and stock assessment models." It presents two complex model equations, which are a bit awkward and somewhat "too detailed" in this brief review.**

*R1_5- I understand the need to balance the in-depth discussion and presentation of each class of models. Thus I removed a lot of detailed equations and explanations, keeping the description for each class of models equally synthetic and leaving more space for discussing the readiness for operational applications, as well as for challenges. In particular, Section 3 was shortened by removing all the equations and detailed explanations.*

**Some technical corrections:**

**Page 2, line 31. Should be "six classes"**      *Ok,done*

**Page 5, line 151. What does "population dynamics under technical interactions" mean?**      *An unnecessary detail: removed.*

**Page 7, line 195. "These WEM models..."**      *corrected*

**Page 15. Table. What is the gww?**      *An unnecessary detail: removed.*

**Page 16. Table. Missing the "of" after the "Number"**      *corrected*

**Page 17. Table. "Ensemble" instead of "Ensamble"?**      *corrected*

**RC2**

This paper provides a classification of numerical models and toolboxes developed in recent years for marine ecosystem modelling with a view to ecosystem management and prediction. Given the limited space, the description is succinct and the overview cannot meet the objective of a full review. The readers will inevitably have to delve deeper, consulting additional reviews, synthesis or original articles to access the information they're looking for. The diversity of models available today has become very large, and the classification proposed here is useful in guiding the user not directly involved in specific model developments.

*R2_0: thanks for the comment. Clear references to other reviews to dig deeper is not explicit in the paper.*

Therefore, the paper should be published but some aspects need to be improved to make the contribution more impactful:

1. **The purpose of the paper and target audience should be clarified and made more explicit. Is it a "review of reviews" (as suggested page 2, line 31), or an attempt at classification, or a user's guide, or something else ?**

*R2_1: The work does not intend to be exhaustive for any of the approaches, thus cannot be considered a user guide. Rather, it provides a classification of a spectrum of approaches broader than each single review considered and cited. Each class is reported with examples of the most used high trophic level tools to help the reader to orientate in the vast set of models. Moreover, specifically the paper report indication of the potential use of each model for an operational connection with lower trophic level models and therefore it can be considered a broad map of ecosystem models for ecosystem operational oceanography. This is now specified in the introduction part*

2. **Is the classification mutually exclusive, or can there be models that fall into several classes, or combine concepts from several classes? A short discussion on possible intersections between classes should be added.**

*R2_2: A short discussion on intersections between classes of models was added in the discussion part. In fact, the classification has some degree of overlap. For instance, the difference between Minimum Realistic Models and*

3. **One of the aims of the paper seems to be the assessment of the "operational maturity" of the models listed. The notion of "operational model" needs to be clarified in the text. Is it a question of high TRL ? of ability to produce information routinely ? in real time ? The title mentions "Numerical Models for Monitoring and Forecasting", but the text suggests that the focus is more on management issues. It would be useful to identify the reasons why some models are still far from operational status, once the notion is clarified.**

*The paper intends to provide a structured synthesis of numerical approaches to connect lower trophic level models (physics- biogeochemistry and plankton) to higher trophic levels. The paper aims at covering - in an organized way - approaches from single individual to population to multispecies and more emphasis might be placed on their utilization as tools to be connected to Copernicus Marine Service products. This is now more clearly stated in the introduction*

4. **The content of the sections on the 6 classes might be harmonized. Equations or GitHub references are used in some sections, but not in others. Sections 6 and 7 do not give any conclusions on the operational maturity (or TRL) of the models, unlike the others. Maybe**

> **provide one example of emblematic model for each class ? I suggest to remove the equations in section 3.**

*Suggestion was followed: The paper content was harmonized, section 3 was much simplified. Moreover, descriprion of maturity and readiness of each model approach was included in each section and in table 1.*

5. **From a user's perspective, it will be very useful/important to identify in Table 1 (i) one reference paper for each model ; (ii) which models are open access, and how to access the code in practice (e.g. on GitHub).**

*Thanks for the suggestion very prepositive, useful for improving the paper. Suggestion was followed and the columns to report reference and open access repository were included.*

**Minor comments:**

**- page 2, line 31: five -> six classes**     *corrected*

**- page 2, line 40: "modelled dynamically or derived from energy requirement": I don't understand the alternative in the sentence.**     *rephrased*

**- page 2, line 43: "clear empirical interpretation": interpretation capability ?**     *rephrased*

**- page 3, line 67: how B differs from Bt ? Ct not in (eq 1)**     *removed completely*

**- page 3, line 79: meaning of SS3, a4a, XSA ? please remove unnecessary acronyms.**     *removed completely*

**j- page 4, line 119: would it make sense to cite ichthyop (Lett 2008) explicitly ?  https://ichthyop.org/ which has been used in a variety of projects ? (at least to be added to Table 1)**     *thanks, it was added*

**- page 5, line 147: please provide the GitHub reference**     *In table 1 was included a column relative to repository. This regards github and other public repository where readers can find the scripts.*

**- page 6, line 174: is DBEM the same as the one cited in Section 5 (species distribution models) ? Is this an example of hybrid class model ?**     *revised.*

**- page 7, line 204: "general improvement of realism but also lower accuracies": this is confusing. What is meant by "realism" ?**     *revised.*

**- Table 1: the legend should be checked and further developed to explain to meaning of each column (e.g. time units ? is it time scale ? "Name" should be "Model" and Model should better be "Acronym" ?)**     *thanks a lot for this suggestion. The Table 1 was revised.*

**- Table 1: a new column indicating open access, web site and (one) reference paper would be very helpful**     *thanks a lot for this suggestion. The Table 1 was revised by including columns for repository and for the references*

**- Table 1: the last row (Apecosm) should be filled in**     *revised, ok completed*

**RC3**

The author reviews current numerical marine modeling approaches, from bioenergetic models for individual organisms to ecosystem-scale models, with a stated focus on understanding and managing marine systems under stress from human activities and climate change. The manuscript attempts the difficult task of briefly assessing the maturity of various modeling approaches while discussing their applications in marine resource management.

The manuscript's assessment of model maturity in the earlier sections (2-4) appears subjective and would benefit from more rigorous supporting evidence and citations. Often, it is not even clear what factor contributed to the assessment. An example is the apparent contradiction in the utility of spatial explicitness and oceanographic forcing for stock assessment models and connectivity models that need to be resolved or better explained: "Although in most cases, stock assessment models are not spatially explicit and do not consider explicitly oceanographic forcings they are routinely used in formal assessments for management and might be considered as ready for operational oceanographic applications." (l. 101) Yet, the text describing connectivity models seems to suggest that adding the missing features makes the models less mature: "Connected with oceanographic variables and spatially explicit these models however, appear less mature for operational applications." Perhaps rather than attempting maturity assessments, I would suggest using the approach taken in Section 5, which focuses on describing actual use cases and operational status.

*R3_1: The reviewer's comment on the maturity assessment is in line with the previous reviewers' comments. Basically, I understand that not only maturity was not clearly defined and measured, but a possible confusion between different aspects such as maturity and use in management became apparent. I suggest defining and adding in the table: a) maturity of the numerical approach, b) readiness for coupling with physical-biogeochemical models. Maturity is defined by the free availability of the code, documentation of the model and applications, test cases, diagnosis of model performance, and the reference community that updates and contributes to the model: this is a fairly objective assessment that can easily be included in the table and discussed to facilitate dissemination of the approach. The readiness for physical-biogeochemical coupling are based on capability of the tool to produce large-scale analyses, integrated with lower trophic level models that can assimilate real data and predict marine resource dynamics in space and time. This should contribute to greater coherence, pragmatism and objectivity, although there will still be a degree of subjectivity in the assessment of readiness to monitor and predict.*

In general, I would suggest revising each section to follow a consistent format: beginning with a brief description of the model class, followed by current use cases -- if available -- specifically related to human activities and climate change. Each section could conclude with an assessment of which current environmental challenges are already being addressed or need attention. This would be more valuable than the current approach, which primarily describes existing models and their structure. The list of models in Table 1 already serves this purpose, allowing the main text to focus more on the present status and outstanding issues in the field.

*R3_2: In agreement with other reviewers, there is also a need for RC3 to have a consistent format for the different model classes. An explanation of the current approach, minimal descriptions and applications is now reported in brief, but I think they absolutely need to be included. However, as the reviewer suggests, I improved the description also harmonizing this structure for all classes by*

*shortening some classes description, reducing to maximum 1-2 examples of tools and describing the applications and availability of the code .*

**In Section 2, the description switches from a general description of bioenergetic models to the DEB subclass. If all current bioenergetic models are DEB models, this should be pointed out in the manuscript. If not, what are the alternatives and when are they used?**

**Starting from the title to about the last sentence of the abstract, a reader might think that the topic of the manuscript are coupled physical-biogeochemical models, as these are often referred to as ecosystem models. Here it would be useful to differentiate the type of model earlier, preferentially in the title.**

*R3_4: The title was given. What was done was trying to explicit since the beginning that the content of the paper regards ecosystem models (from physics to fish).*

**In this context, it would be helpful to more clearly state what kind of organisms are being modeled using the approaches. Neither the abstract nor the introduction explicitly mention what is being modeled: The abstract speaks of "individuals, populations, communities and ecosystems", the introduction mentions "marine biological resources" without providing any examples. As stated above, I would suggest making use cases of each type of model more explicit.**

*R3_5: The solution I propose is to start the paper with a better description of the objectives. Namely, the aim of the paper is to provide a structured synthesis of models for the higher trophic levels of the oceans (essentially from zooplankton to top predators and fisheries) that can be linked to models for the lower trophic level (physics and biogeochemistry). The lower trophic level models currently provide operational products in the Copernicus Marine Service, and their potential to be linked to upper trophic level models opens up a wide range of new CMS products. The aim is to cover most high trophic level approaches, from single individuals to populations and multispecies, but the work may not be exhaustive. Conversely, the paper provides a classification and examples of the most commonly used high trophic level tools with some indication of their potential use for operational linkage to lower trophic level models.*

**# specific comments**

**L 22: "numerical models can be divided into six broad classes [...] These five classes of models are reviewed in the following sections ...": It should be six classes in both cases.**
      *revised.*

**L 51: the "DEB" abbreviation was introduced in the preceding sentence.**       *revised.*

**L 53: "abstract concepts that are more challenging to measure empirically": What would be an example of that?**    *The abstract has been revised.*

**L 54: "The presence of the storage": Here it would be easier for the reader to refer to "storage tissue" again, using the exact term introduced earlier.**    *revised. The intention here was to point out the "storage" of the DEB modelling approach, which is quite difficult to measure empirically. However, I concede that it is not an abstract concept. The text was revised here.*

**Eq 1: What is n here?** *This equation has been removed*

**L 67: "fishing mortality": So this type of model is only used to model populations of species that are being fished? Why not mention this early on explicitly?** *revised.*

**L 79: "such as SS3, a4a, XSA, etc.": Listing these names/abbreviations here is not useful. Readers unfamiliar with these approaches are only given 3 letter abbreviations without any context or citations, and readers who know these approaches probably don't need the see the abbreviations again. Here, it would be much more useful to describe these sophisticated approaches in words.** *Lot of this description has been removed for the sake of simplicity. Moreveor the descriptions of the six classes has benne made more uniform.*

**Eq. 2: I think it is not useful to show the equation here, it can be more easily explained in words. If it is kept, the initial N should have a subscript t, and the "a=" is missing from some of the subscripts.**

**L 153: "considering the specific problem": What is the specific problem?**

*The intent here was to explain that MICE is a problem-specific model. Basically it is adapted to describe the ecosystem on the basis of the question to answer or issue to assess. This text was revised for increase clarity.*

**L 157: "MICE can simultaneously represent focal populations in age-structured classes, while others take a surplus production approach": This could be described better, what is the surplus production approach, and how does it compare to an age-structured approach?**

*agree, more details and explanations were needed in order to allow non expert of this kind of models to appreciate their characteristics. I removed from here the text and simplified a lot the description.*